# A dataset of human and robot approach behaviors into small free-standing conversational groups

**Fangkai Yang**[1]*, **Yuan Gao**[2◉], **Ruiyang Ma**[1◉], **Sahba Zojaji**[1], **Ginevra Castellano**[2], **Christopher Peters**[1]

**1** Department of Computational Science and Technology, KTH Royal Institute of Technology, Stockholm, Sweden, **2** Department of Information Technology, Uppsala University, Uppsala, Sweden

◉ These authors contributed equally to this work.

* fangkai@kth.se

**Data Availability Statement:** The data underlying this study are available on Zenodo (https://zenodo.org/record/4537811).

## Abstract

The analysis and simulation of the interactions that occur in group situations is important when humans and artificial agents, physical or virtual, must coordinate when inhabiting similar spaces or even collaborate, as in the case of human-robot teams. Artificial systems should adapt to the natural interfaces of humans rather than the other way around. Such systems should be sensitive to human behaviors, which are often social in nature, and account for human capabilities when planning their own behaviors. A limiting factor relates to our understanding of how humans behave with respect to each other and with artificial embodiments, such as robots. To this end, we present *CongreG8* (pronounced 'con-gre-gate'), a novel dataset containing the full-body motions of free-standing conversational groups of three humans and a newcomer that approaches the groups with the intent of joining them. The aim has been to collect an accurate and detailed set of positioning, orienting and full-body behaviors when a newcomer approaches and joins a small group. The dataset contains trials from human and robot newcomers. Additionally, it includes questionnaires about the personality of participants (BFI-10), their perception of robots (Godspeed), and custom human/robot interaction questions. An overview and analysis of the dataset is also provided, which suggests that human groups are more likely to alter their configuration to accommodate a human newcomer than a robot newcomer. We conclude by providing three use cases that the dataset has already been applied to in the domains of behavior detection and generation in real and virtual environments.

A sample of the CongreG8 dataset is available at https://zenodo.org/record/4537811.

## Introduction

A typical human interaction pattern in natural environments is the formation of small groups of individuals that gather and stand together to converse. These social formations, referred to as *free-standing conversational groups* [1], are a common means by which individuals naturally

**Funding:** Grant Number: 765955 Grant Recipients: S.Z., G.C., C.P. Funder Name: H2020 European Institute of Innovation and Technology The funders had no role in study design, data collection and analysis, decision to publish, or preparation of the manuscript.

**Competing interests:** The authors have declared that no competing interests exist.

interact and collaborate in situated contexts. Thus, deepening our understanding of them can support efforts for creating safe, efficient and effective collaborations between humans and artificial systems. An especially important phase in such interactions relates to the situation when an individual, or *newcomer*, approaches a group with the intention to join it. While such an event may seem trivial at first glance due to the effortless nature with which we appear to be able to conduct it, a deeper inspection reveals hidden intricacies relating to a subtle exchange of non-verbal signals that lead to a group making space for a newcomer by accommodating it, or ignoring it forcing the selection of an alternative strategy that may risk interruption or a loss of *face* [1] for participants. In addition to such social considerations, the basic planning problem is also not trivial. As a newcomer approaches a formation, the positions and orientations of group members may change, requiring replanning of approach trajectories if the newcomer wishes to be seen. Any comprehensive study of these phenomena must therefore note their dynamics, accounting for the trajectories of those approaching a group to join it (Fig 1) and the reaction of group members. Such a task is especially challenging for artificial systems, such as mobile robots, which may find it difficult not only to perceive the environment, but able to understand these cues [2, 3] to better predict if a group is changing formation to accommodate it. These predictions are a basis for planning safe trajectories into a group in a socially-acceptable manner [4, 5], so that a variety of undesirable consequences do not occur, such as the system interrupting the group unnecessarily or even colliding with one of its members.

Recent research works have focused on simulating such behaviors [4, 6] from the perspective of robot learning using either prior models or synthetic data. However, it is unclear how the resultant approach behaviors from such models perform in actual human-group and robot-group interactions due to the lack of group interaction datasets. Those existing datasets that contain free-standing groups [7–9] have a limited number of samples of individuals approaching the group and typically contain only 2D location information, making it difficult to train neural networks. To overcome these difficulties and better reveal interactions at the group level, this paper describes *CongreG8*, a novel dataset consisting of human-group interaction data, robot-group interaction data, personality data, and custom human/robot interaction questionnaires. This paper is strutured into three main parts. In the first, an overview of the dataset is presented. An analysis of the behaviors and associated questionnaires is then presented, suggesting that small groups are more likely to accommodate a human newcomer than a robot newcomer. This behavior does not appear to significantly relate to the personality of participants. To conclude, we present three use cases to demonstrate the utility of our dataset in a variety of domains: group behavior recognition, robot behavior generation, and the animation of small group behaviors.

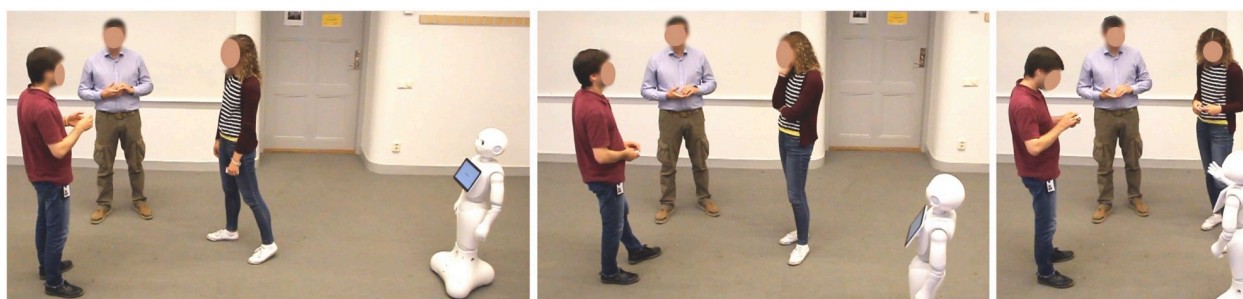

**Fig 1. A representative image of the encounters captured in the CongreG8 dataset.** In this case, a Pepper robot approaches a free-standing conversational group in order to join it. The complete dataset consists of trials of human approach behaviors and robot approach behaviors. Actual recordings took place in a motion capture facility in which all participants (apart from Pepper) wore motion capture suits.

To our knowledge, this is the first full-body motion capture (*mocap*) dataset focused specifically on approach to join behaviors for small groups, and also the first to include robot approach behaviors in addition to solely human-group data. Moreover, CongreG8 aims to promote standardization and benchmarking in human-robot interaction (HRI) research, by providing, for the first time, a benchmark against which to compare different computational methods for the automatic classification of group behaviors in HRI and learning how a robot should approach groups. This is of the utmost importance to enable comparability and reproducibility of results in HRI research.

## Related work

### Group interaction research

There have been numerous studies on group interaction, with fewer focused on situations in which a newcomer approaches and joins a group. In a free-standing conversational group, Kendon [10] proposed the *F-formation* system to define the positions and orientations of individuals within a group. F-formations and other group formation models have been studied computationally [11–15] with potential applications to mobile robots [16] and wheelchairs [17], and have been used as a basis for joining group behaviors of a mobile robot [18–21]. Truong et al. [18] proposed a framework that enables a robot to approach a group safely and socially. Escobedo et al. [17] infers joining group destinations by considering contextual information and user's intention. Other models [19, 22] extended a fast marching algorithm to navigate a robot for engaging a group of people. Althaus et al. [20] developed a topological map-based model for a robot to approach a human group. Other works [23, 24] focus on investigating the factors that affect the perception of a conversational group towards robot behaviors, such as the distance and angle at which the robot approaches when joining a group. In the virtual environment, Pedica et al. [25] augmented the Social Forces model [26] for simulating virtual character's joining group behaviors. Attractive and repulsive forces were used to drive towards the target and conduct collision avoidance. All aforementioned works are either experimental studies or computational models implemented and validated in a simulation that rely on manually specified features. Other recent works have made use of data-driven methods concerning joining group behaviors [4, 6, 27, 28]. However, they were trained using synthetic datasets or prior computational models due to the lack of real-life datasets. All of the research described in this section could benefit from the CongreG8 dataset, whether to inform or help validate manual features or to be used directly in data-driven approaches.

### Group interaction datasets

Human-human interaction databases were extensively reviewed in several surveys including [29–31]. Unlike datasets containing individual action recordings, human-human interaction datasets, i.e., those containing multiple humans interacting, are relatively scarce. One is the CMU Panoptic Dataset [32]. In this dataset, different kinds of interactions, such as dance and haggling, are collected. The advantage of this dataset is that the recordings are relatively accurate, although a disadvantage is that the recording space is limited if trajectories are to be considered. Another dataset is the BARD dataset [33]. With a focus on human behavior analysis in video sequences with multiple targets, the dataset records human interactions in wild environments. However, there is no particular joining behavior in these scenarios. In addition, a recent dataset MHHRI [34] focuses on analyzing and comparing the natural behavior of human-human and human-robot interactions. Although it contains trajectories of head and hand movement, group approach behaviors are not considered. The SALSA dataset [7], the MatchNMingle dataset [9], and the Idiap Poster dataset [35] contain a limited number of

group approach behaviors with only position and orientation information. CongreG8 contains more group behaviors with detailed 3D full-body information that could be used to train and understand group behaviors. Other recent datasets focus on egocentric vision information. For example, JRDB [36], RICA [37], and RoboGEM [38] contain collected videos and images of human crowds from egocentric mobile robots. While these datasets are useful for training systems from an egocentric perspective, the CongreG8 dataset provides a more global view of group interactions. Since it is based on motion-captured data, it offers high quality 3D full-body information with fewer occlusions and higher continuity.

## Materials and methods

### Data collection scenario

In order to provide structure to the group interaction behaviors, a game *Who's the Spy* was designed as the scenario. This game involves three players, positioned in a *conversational group*, and one player (or alternatively, one robot) in the role of the *adjudicator* (Fig 2). In every game round, each player in a group is given a card with a word on it. Among them, two cards have the same word, while the third one has a different word. The player who has the card with a different word is the spy. Players are only able to see their own cards. When the game starts, the players take turns to describe the objective properties of the word they have in hand. The adjudicator has the role of identifying the spy and thus does not see the cards of the

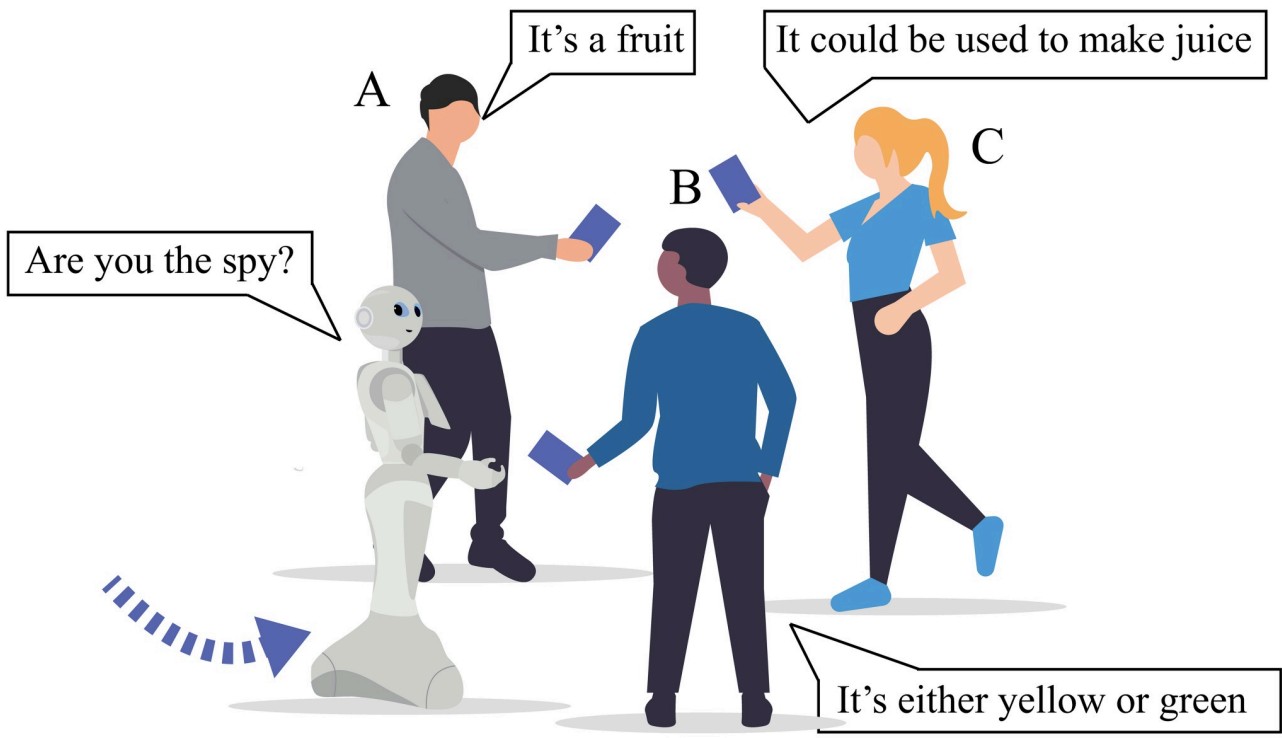

Card Words: A-Apple, B-Lemon, C-Apple

**Fig 2. The *Who's the Spy* scenario.** Each of the three players in the conversational group has a card with a word on it. Only one word card is different. The player who holds the odd card is the spy, but players do not know what cards the others have. They take turns to describe the word until the adjudicator (either a robot or a human) approaches the group to identify the spy. In this example, player B is identified as the spy and all the players show their cards to confirm that the identification is correct.

players. They stand 1-2 meters outside of the conversational group and observe the ongoing conversation. Once the adjudicator establishes the identity of the spy or the time for the round is up (each game round is set to be maximum 1 minute), they approach and join the conversational group in order to inform the players of the outcome.

For example, as shown in Fig 2, three players (*A*, *B*, and *C*) are formed in a conversational group and are holding word cards with the words *Apple*, *Lemon*, *Apple* written on them respectively. Clearly, player *B* is the spy, but none of the players (including player B) or the adjudicator knows it. Player *A*, who has the word *Apple*, could say *"It's a fruit"*. Then player *B*, who has the word *Lemon*, could say *"It's either yellow or green"*, and then player *C* takes the turn, *"It could be used to make juice"*. The conversation progresses, and players are not allowed to repeat previously described properties. In the meanwhile, the adjudicator stands 1-2 meters from the conversational group and monitors the conversation. The adjudicator approaches and joins the group once they identify the spy or when the time is up, whatever case occurs first. The players in the group then display their word cards to the whole group in order to confirm the identification.

The game has been designed to collect the full-body behaviors of players in the conversational group in addition to the adjudicator, which as a newcomer to the group, engages in numerous approach and join behaviors. The players in a group do not know when the adjudicator will approach it to identify the spy and are engaged in the game in the meanwhile. The game therefore simulates situations in which a conversational group does not know whether or when a newcomer will approach the group and attempt to join it. Such approach behaviors may be difficult to capture in natural settings due to the rare relative frequency with which they occur (see publicly available datasets containing conversational groups [7, 35]). The scenario used in our dataset was therefore chosen to represent such a setting while also providing a large number of approach behavior samples in as efficient a manner as possible.

## Experimental conditions

The CongreG8 dataset provides two baselines for behaviors related to approaching and joining groups: a *human-group condition*, in which a human plays the role of the adjudicator, or newcomer, and approaches the group, and a *robot-group condition*, in which a robot plays the adjudicator/newcomer role.

**Human-group interaction.** In the human-group condition, we expect to observe natural and diverse approach behaviors from the human newcomer and the corresponding group behaviors when reacting to the newcomer. From the perspective of machine learning in general, we collected more human-group interaction data to provide a training dataset for learning approach group policies and recognizing group behaviors [39].

Three small booklets of word cards are distributed to three group players, and each booklet contains 40 word cards with an order that ensures only one different but synonymous word exists in one game round. The group players are not directed to stand on any particular position, but they are instructed to stand freely around the room center for a better motion capture quality (Figs 3 and 4). When the game starts, each player takes turns to play as the adjudicator and alternates after 10 game rounds. For instance, player *D* acts as the adjudicator and player *A*, *B* and *C* stay as a group in the first 10 rounds, then player *C* hands over the booklet to player *D* and takes the place of player *D* by playing as the adjudicator for the next 10 rounds.

Prior to the adjudicator approaching the group, they stand outside of the group area but are still within the camera capture area. The human adjudicator is instructed to walk around the room before approaching and join the group. The purpose is to prevent the adjudicator from always approaching the group from the same direction. From our pilot experiments, if the

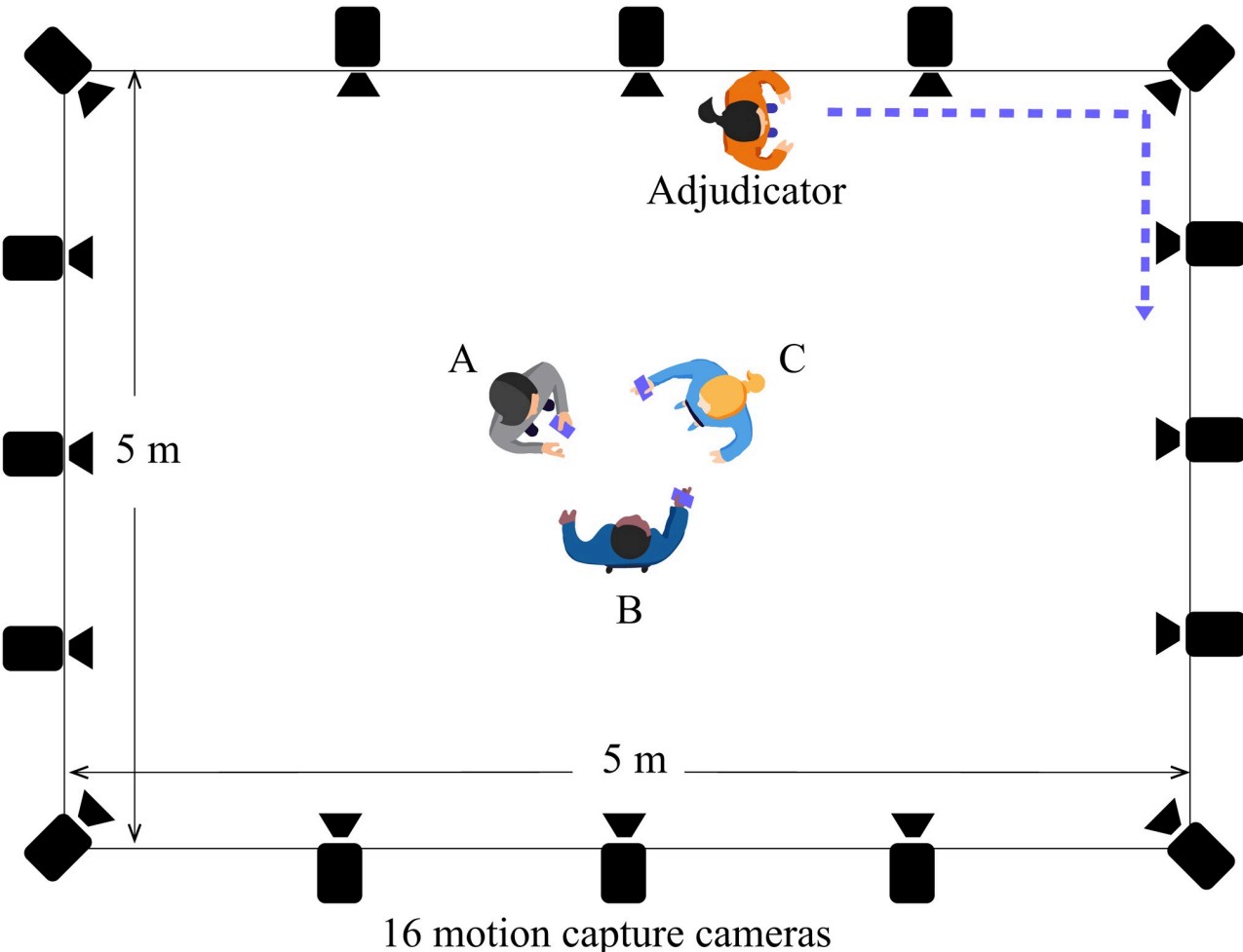

**Fig 3. The human-group condition.** The top-down view of the human-group interactions (room is not to scale). Three players were directed to stand in the center of the room and could do so freely (i.e., were not assigned specific positions). In an attempt to create a more natural situation with a variety of approach directions, the adjudicator was instructed to walk around the periphery of the group and, when directed to do so, approach them.

adjudicator stands still before approaching the group, the group members tend to orient their upper-bodies to make space for the adjudicator before being approached. Feedback after the pilot experiment suggests that the group players are aware of the approaching direction, and they know the adjudicator will approach from a specific direction later. In addition, if the adjudicator is placed in specific positions and stands still, the group may be not aware of them. Hence walking around the room makes the adjudicator's approach direction more random with respect to the group members.

**Robot-group interaction.** In the robot-group condition, the human adjudicator is replaced by a physical Pepper (https://www.softbankrobotics.com/emea/en/pepper) robot. This condition provides a baseline for the social interactions and dynamics between an approaching group robot and a conversational group, in the meanwhile, keeps a similar social setting as the human-group condition.

An experimenter remotely controls the robot via a Wizard of Oz (WoZ) approach [40, 41]. The details of robot control will be described later. Players do not know that an experimenter is controlling the robot while they are playing the game, and they are told the robot is fully

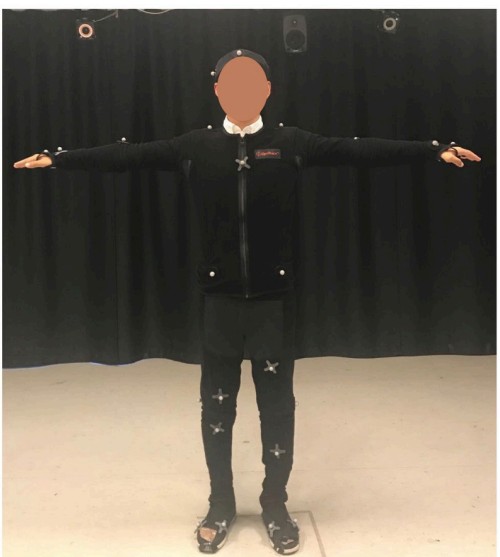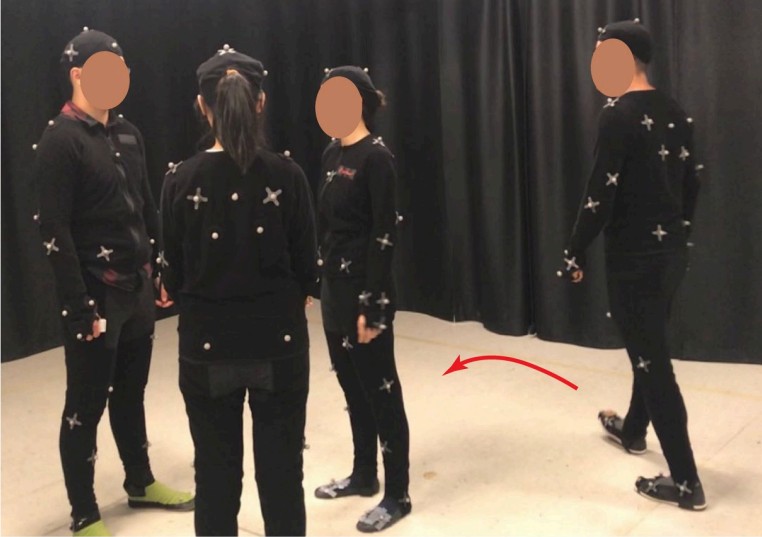

**Fig 4. The human-group condition.** Setup for the human-group interaction motion capture. A participant in a motion capture suit, T-pose (left). A group of three play *Who's the Spy*, and the adjudicator approaches to join them (right).

autonomous. Similar to the behaviors of the adjudicator in the human-group condition, the robot initially stays outside of the group but within the capture area before each trial starts, and it walks around the room before approaching the group. Once the experimenter has determined the spy, or this round time is up, they control the robot to approach and join the group. During this phase, face-tracking is enabled on the robot so that it orients its head towards a player and asks if they are the spy. The players then show their cards to confirm if the robot is correct, at which point it provides a verbal response and accompanying pre-programmed postures and gestures (see Fig 5 right).

## Participants

Forty participants (27F:13M) aged between 22 and 35 (M = 25.8, SD = 3.2) were recruited from the university locale at KTH Royal Institute of Technology through public bulletins and online advertisements to participate in the motion capture sessions. The 40 participants were randomly divided into 10 *participant pools*. They were not allowed to choose which pool they would go to in order to reduce situations in which previous acquaintances would decide to join the same pool. Within each pool, the roles of newcomer (adjudicator) and conversational group member were rotated throughout the session, as mentioned in Section *Data collection scenario*. All 40 participants took part in the human-group interaction session and a subset of 16 participants took part in the robot-group interaction session. Each participant was compensated with a cinema e-ticket for their time.

## Hardware

Motion data was recorded in a motion capture lab with an approximate 5m × 5m × 3m active capture volume, which is equipped with a NaturalPoint Optitrack (https://optitrack.com/) system with 16 Prime 41 cameras. Each camera has a 4 megapixel resolution with a frame rate of 120 fps. The motion of each human player was recorded with a Motion Capture suit with 37 markers (Fig 6) placed at respective anatomical locations of the body (see Fig 4 left). These markers are attached to the surface of the body in order to capture full-body behaviors. On the

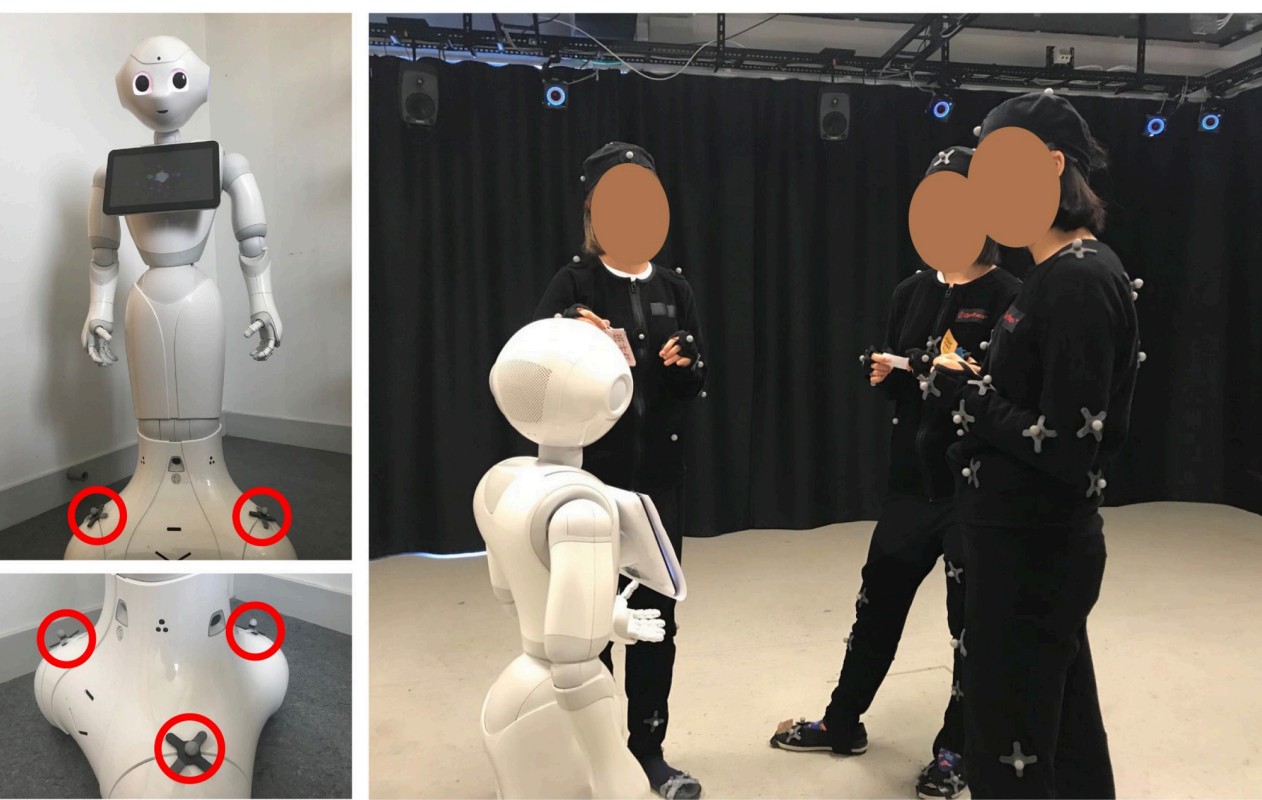

**Fig 5. The robot-group condition.** Setup for the robot-group interaction motion capture. The robot has 3 markers (red circles) attached on the base in order to track its position and orientation (left). The robot acts as the adjudicator to join and find the spy (right).

other hand, the motion of the robot, including its position and orientation, was recorded from 3 markers attached to its base (see Fig 5 left).

## Software

The motion capture process is managed by Motive (https://optitrack.com/products/motive/), motion capture software designed for both capturing and processing of 3D data and 3D information reconstruction from live-streamed data. In the robot-group condition, the experimenter controls the robot remotely through a Python script developed using Naoqi SDK (http://doc.aldebaran.com/2-5/index_dev_guide.html) and Pygame APIs (https://www.pygame.org/wiki/GettingStarted) (the script is shared together with the dataset). In addition, the experimenter use both the camera view from the robot forehead camera from Choregraphe (http://doc.aldebaran.com/2-4/software/choregraphe/index.html) and the reconstructed skeletons from Motive to better perceive the real-time experimental environment remotely when con-trolling the robot (Fig 7).

## Robot control

In the robot-group condition, the adjudicator is replaced by a Pepper robot controlled by an experimenter via a WoZ approach. A WoZ approach was adopted since human control was the most simple, reliable and robust method for moving the robot into an appropriate position, given the dynamics of the scenario and group situation. The experimenter remotely controls the robot through a Python script via a keyboard. Four keys are used to control the left/right/

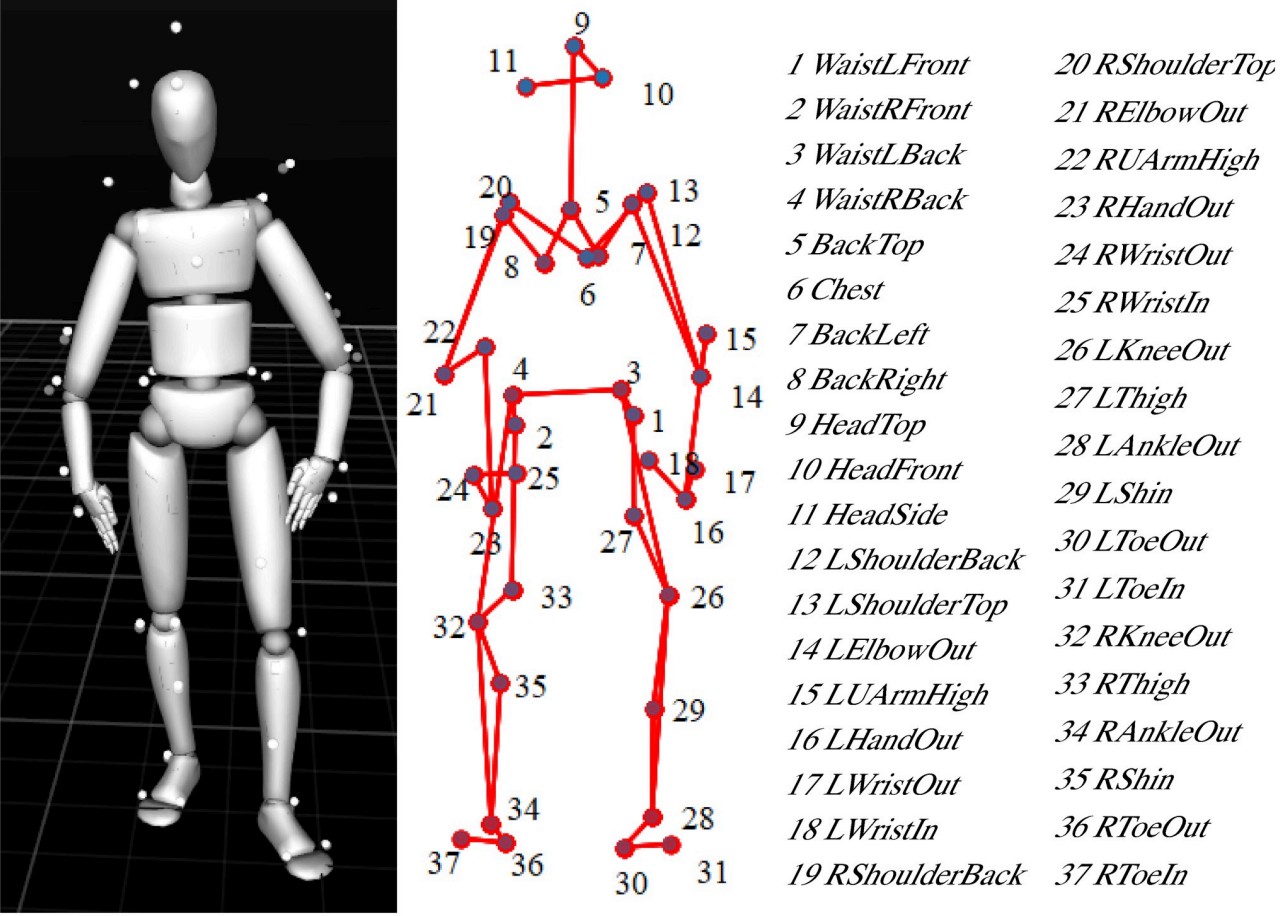

**Fig 6. 37 full-body markers.** Reconstructed skeleton (left). Marker positions (middle) and names of 37 markers (right).

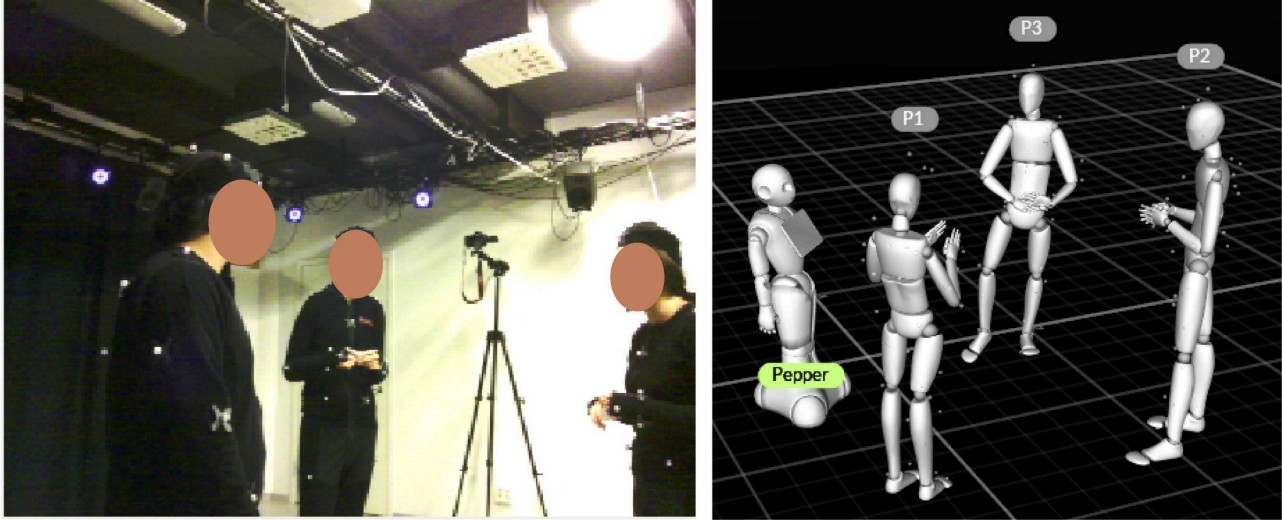

**Fig 7. Real-time views that help the experimenter to control the robot.** The camera view from the robot's forehead camera (left). The reconstructed scene from Motive including three group players and the robot (right).

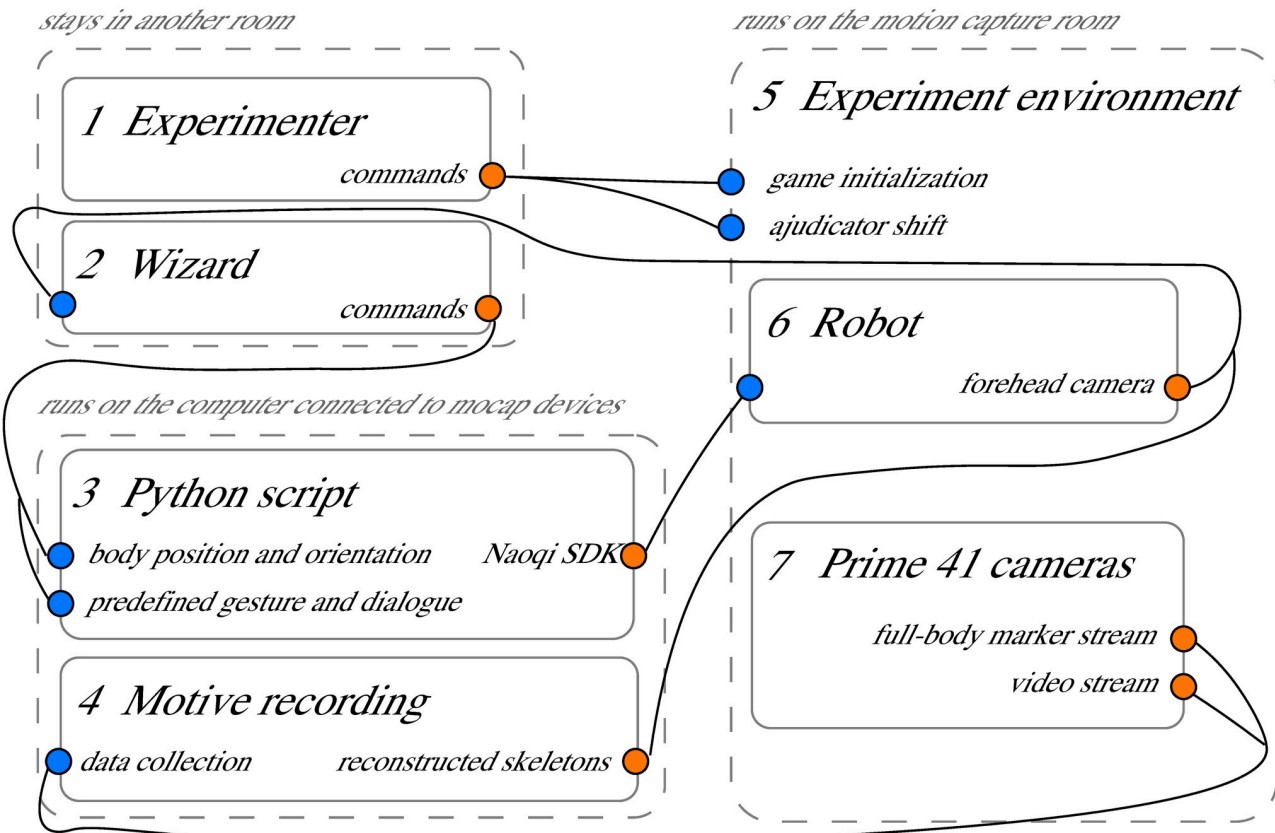

**Fig 8. Diagram of the protocol for the data collection scenario.** The data or stream flows from orange dots to blue dots. In the robot-group condition, the wizard (2) controls the robot through a Python script (3), including body position and orientations. In addition, the robot (6) presents predefined gestures and dialogues when it identifies the spy after joining the group. Real-time constructed skeletons from Motive (4), and the forehead camera view from the robot are used to help the wizard send robot control commands.

forward/backward movements of the robot, and two keys are used to control the left/right turning. As described in the software apparatus section, Fig 7 shows both the camera view from the robot's forehead camera and the reconstructed 3D information from Motive. They are used to help the experimenter better perceive the real-time environment remotely. However, in order to ensure that participants would treat the robot as an independent agent rather than an avatar representing a human, participants were informed that the robot was fully autonomous. Fig 8 presents the architecture of the scenario with the data flow.

### Protocol

The data acquisition protocol is presented in Table 1. In the *Greeting* stage, the participants are told that the robot is fully autonomous. In the robot-group condition, after the robot joins the group, predefined gestures and dialogues are triggered when the robot conducts the identification. In the meanwhile, the built-in face tracker of the Pepper Robot is enabled in order to check the spy. These behaviors demonstrate the robot's social capabilities to participants and highlight its potential, beyond its embodiment, as a social entity.

### Data collection

The raw data collected during the experiment is stored in a file format .*TAK* (a Motive file format). Each game round is stored as a single TAK file, which contains all the information

**Table 1. Data acquisition protocol.**

**Greeting (about 15 min)**

- explain the general purpose of the study and the data collection procedure.
- briefly present the Pepper robot.
- ask the participants to fill a consent form and inform them that they could stop their participation at any time.
- ask the participants to complete BFI-10 [42], a short version of the *Big Five Inventory* [43] which evaluates five traits assumed as constitutive of personality.

**Initialization (about 15 min)**

- the experimenters help each participant to wear a Motion Capture suit.
- the experimenters operate the motion capture software *Motive* to reconstruct a skeleton from full-body markers of each participant. The markers are adjusted as necessary by the experimenters to ensure they are placed in correct positions.
- each participant is given a unique ID, i.e., A, B, C, and D.
- three small booklets containing 42 words each are distributed to A, B, and C.

**Tutorial (1-2 min)**

- explain how to play the game, ensure the participants are aware of the game rules. The first two word cards from each booklet are used in the tutorial.

**Human-group interaction (about 50 min)**

- three new booklets containing 12 cards each are distributed. In the first 10 rounds, player D acts as the adjudicator. After the 10th round, player C hands over his/her booklet to player D and replaces player D as the adjudicator.
- after the 20th round, player B hands over the booklet to player C and acts as the adjudicator. Repeat until 40 rounds are done.

**Break (about 10 min)**

- ask the participants to take a break and fill in a Human Study questionnaire.

**Robot-group interaction (about 15 min)**

- in the first 3 rounds, player A, B, and C stay in a group, and the robot always acts as the adjudicator. In the next 3 rounds, player D replaces player C.

**Debriefing (about 10 min)**

- ask participants to fill in a Robot Study questionnaire and a Godspeed Questionnaire Series (GQS) [44] questionnaire. Collect motion capture suits and booklets.
- ask for feedback, answer all possible questions, and give cinema tickets as rewards.

necessary to recreate the entire capture from the file during the whole game period. The time limit for the adjudicator to identify the spy in each game is 1 min, and it results in about 1 min interaction (including final identification and confirmation), and each TAK file is thus about 3 GB. Besides the TAK files, the calibration files are also included in the CongreG8 dataset to support the reconstruction of the motion capture settings.

## Data post-processing

The post-processed data is presented in Table 2. The approaching group behaviors are the focus of this experiment. The raw data is thus post-processed to extract the period from when the adjudicator starts approaching the group to the time they join the group, and the approach behaviors last around 2-6 seconds.

There exist tracking errors in the raw data, including marker occlusions and labeling errors. Marker occlusions result from losing track of certain markers in some frames. These missing markers could be occluded by participant's arms or being out of motion capture area, and they introduce gaps in the data trajectory. For these occluded markers, Motive is used to make interpolations to model the occluded trajectory using the captured data of occluded markers

**Table 2. List of post-processed data in the CongreG8 dataset.**

| Domain | Type | Details |
|---|---|---|
| human | full-body | 37 markers per person (3D position) 21 reconstructed bones per person (3D position and 4D Quaternion rotation) |
| robot | base | 3 markers on the robot base (3D position) 1 reconstructed rigid body (3D position and 4D Quaternion rotation) |
| annotations | | annotated human-group behaviors and robot-group behaviors |
| questionnaires | pre-study | BFI-10 [42] |
| | post-study | Human Study questionnaire Robot Study questionnaire Godspeed Questionnaire Series (GQS) [44] questionnaire |

(see Fig 9 left). On the other hand, labeling errors include unlabeled markers, mislabeled markers, and label swaps. These errors cause incorrectly reconstructed skeletons (see Fig 9 right). Labeling errors are fixed in the data post-processing by assigning appropriate labels to those markers manually.

## Dataset

The CongreG8 dataset (see S1 Table for overview) contains data of human-group interactions and robot-group interactions. The data collection took place in PMIL motion capture lab at KTH Royal Institute of Technology over a period of 1 month.

The CongreG8 dataset contains 380 human approach trials and 38 robot approach trials after data post-processing. Corrupted trials were discarded. Each trial includes full-body motion capture data of all players and the robot (if it is used) during a time period of 2-6

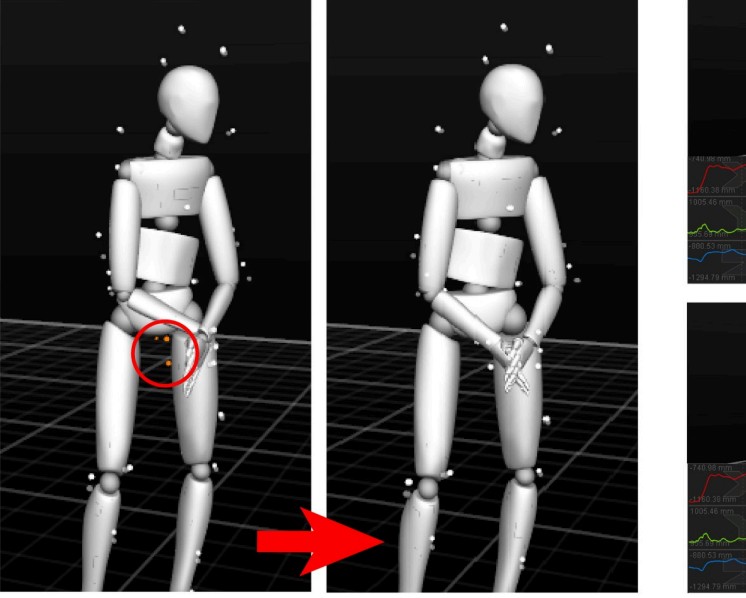 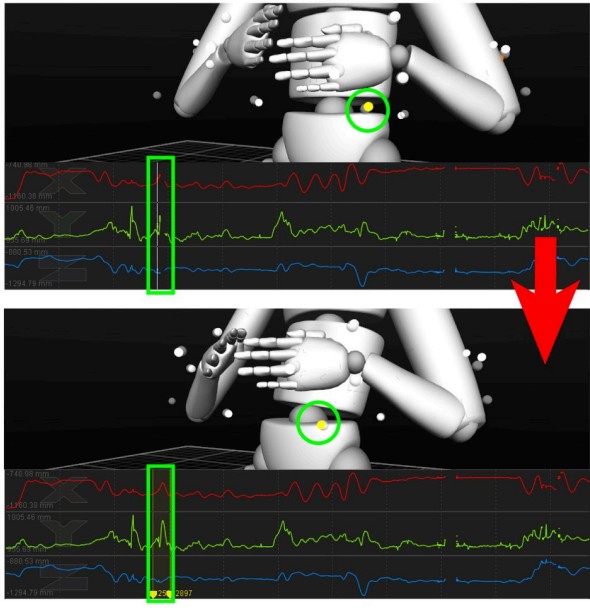

**Fig 9. Data post-processing including fixing labeling errors (left) and marker occlusions.** The three orange markers in the red circle represent unlabeled markers, and we manually assign correct labels to these markers in order to reconstruct the correct right-hand skeleton. The yellow marker in the green circle represents the occluded marker which causes a gap in the data trajectory (inside the green rectangle), and we make cubic interpolations to fill this gap.

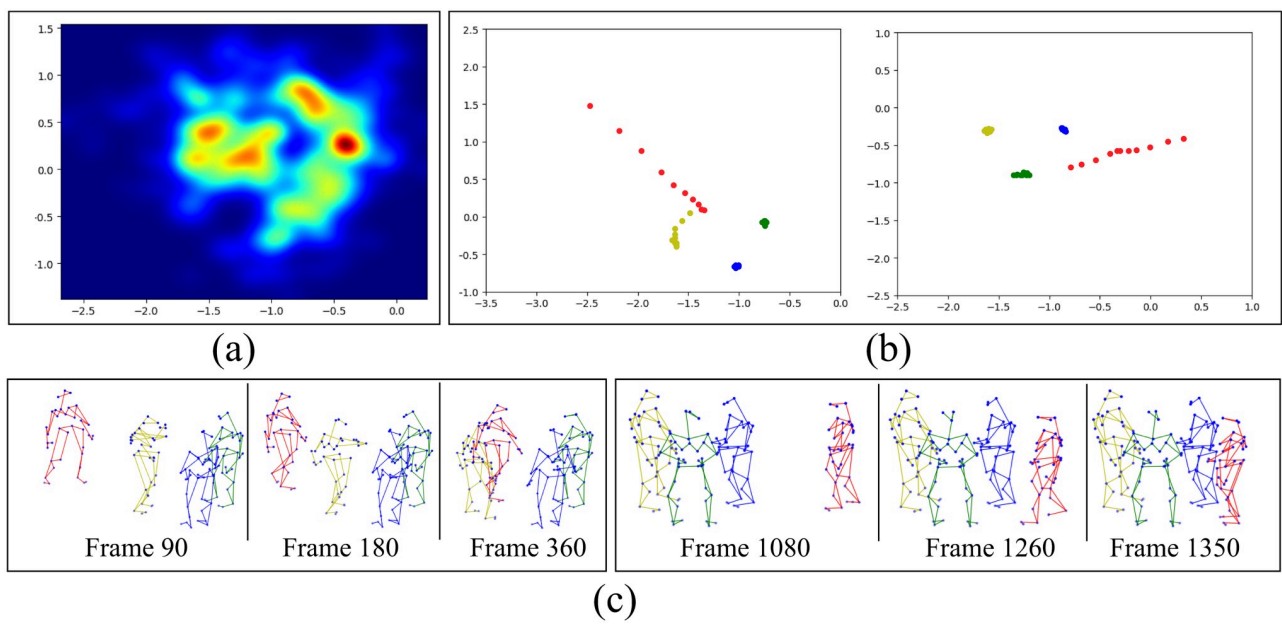

**Fig 10. The visualization of group space and newcomer joining behaviors.** (a) The heatmap of all group members' positions relative to the group center. (b) Two examples of joining group trajectories (the top-head marker is used to represent the position). (c) Full-body markers plots of joining behaviors corresponding to the examples in (b) respectively.

seconds with a frame rate of 120 fps. The data is exported as Comma Separated Values (CSV) files. This file format uses comma delimiters to separate multiple values in each row, and it can be imported by spreadsheet software or a programming script. As shown in Table 2, each CSV file contains 3D positions of all corrected markers and 3D position and rotation (in quaternion format) of the reconstructed skeletons or rigid bodies (the robot). The data is post-processed in order to correct mislabelled markers and replenish missing markers manually. The ConpgreG8 dataset also includes FBX formatted data, a popular format used in 3D animation systems and motion study applications, and gender information could be queried from captured motion.

In the CongreG8 dataset, the group radius has an average size of 0.82 meters (see Fig 10(a)). Fig 10(b) and 10(c) show two randomly selected examples from the dataset to give a general impression of the joining group behaviors. The left images of Fig 10(b) and 10(c) present the *Accommodate* behaviors that group members make space for the newcomer to join, e.g., the yellow group member moves backward to make space. On the other hand, the right images of Fig 10(b) and 10(c) present the *Ignore* behaviors that group members stand still and continue the conversation.

## Annotations

Labeling the behaviors of the participants in the experiment is a complex and challenging process. However, in both human-group interaction session and robot-human interaction session, we found the group displays two general types of behavior, *Accommodate* and *Ignore* (see Fig 11). Three researchers performed the dataset annotation. Each annotator performed the annotation independently, and the final annotation is done by majority voting. The annotators annotated the reconstructed skeleton motions following the definitions (see Table 3). An inter-coder agreement was found to be 91.8% for the human-group trials and 92.1% for the robot-group trials. The group members were not asked to do these behaviors throughout the

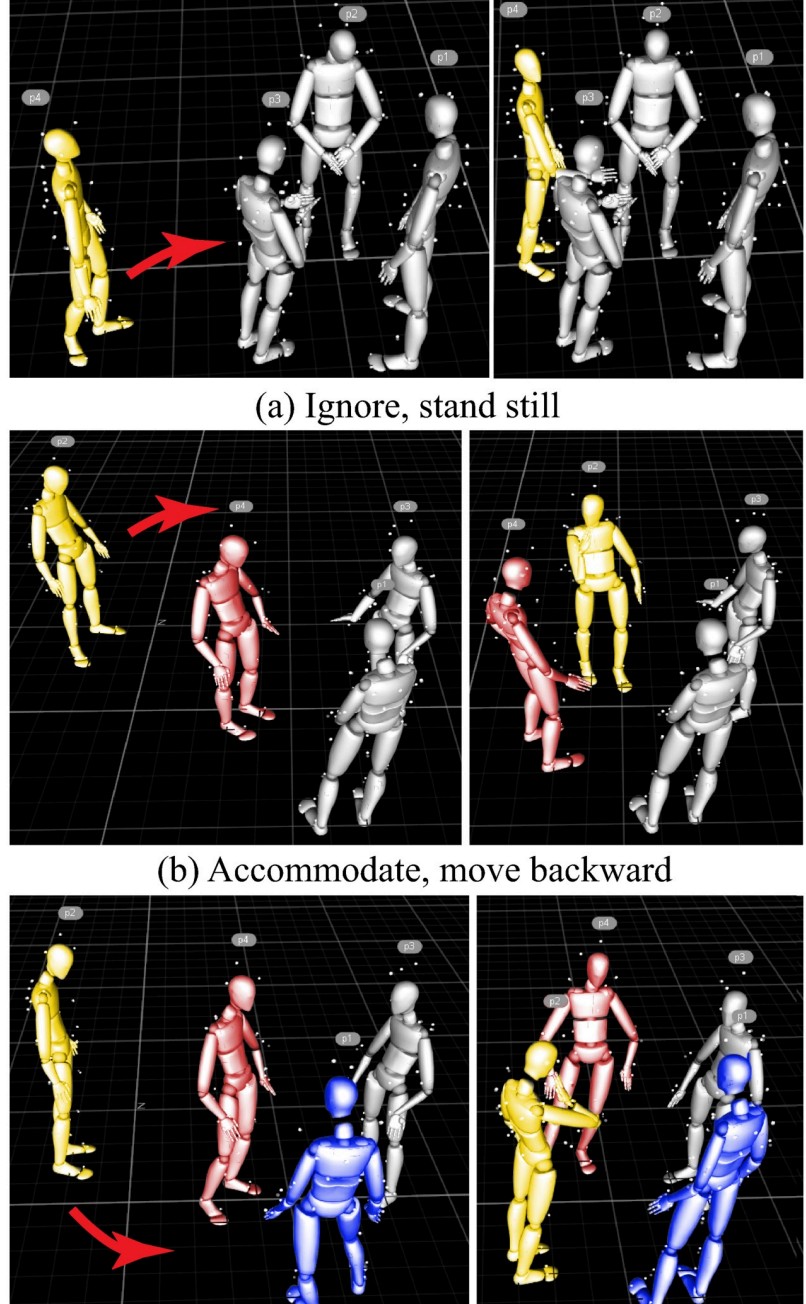

**Fig 11. Two group behaviors when the adjudicator (yellow character) approaches to join the group.** The red arrow indicates the movement of the adjudicator. (a) The group members stand still and ignore the adjudicator purposefully. (b) The group members accommodate the adjudicator, with one group member (red character) moving backward in order to make space for them. (c) The group members accommodate the adjudicator, one group member (red character) moves backward, and another (blue character) shifts weight from one foot to the other. These behaviors make space for the adjudicator.

**Table 3. Group behavior label definition.**

| Type | Definition |
|---|---|
| Accommodate | Group members orient upper-body and eye gaze towards the newcomer, shift weight between feet and/or move backwards in order to make space. |
| Ignore | Group members continue the group conversation regardless of the newcomer and/or stand still. May also glance at the newcomer. |

experiment. In addition, the participants are randomly assigned to each group. It is unlikely that the *Accommodate* behavior comes from the reason that the participants were highly acquainted. The *Ignore* behaviors cannot be interpreted as a lack of knowledge that the group members have the ability to perceive the approaching adjudicator is there, which means the *Ignore* behaviors are on purpose.

Importantly, CongreG8 provides both raw data and post-processed (corrected) data. Researchers who are interested in extracting alternative labels can conduct new annotations based on their annotation schemes, for example, on within-group behaviors before the newcomer approaches.

## Questionnaires

The participants were asked to complete a pre-study questionnaire and three post-study questionnaires. All these questionnaires are included in the dataset.

The pre-study questionnaire is a BFI-10 [42], a short version of the *Big Five Inventory* [43] which evaluates five traits assumed as constitutive of personality: *Extraversion-* being outgoing, energetic vs. solitary, reserved; *Agreeableness-* being friendly and compassionate vs. challenging and detached; *Conscientiousness-* being efficient, organized vs. inefficient, careless; *Neuroticism-* being secure and confident vs. sensitive and nervous; and *Openness-* being inventive, curious vs. consistent, cautious. In the questionnaire, each trait is investigated via ten items assessed on a 1-7 Likert scale. The trait scores were calculated by the detailed procedures in [45]. Fig 12 shows the distributions of these traits over the 40 participants. S2 Table summarizes the distribution of personality and group behavior labels.

The post-study questionnaire consists of a Human Study questionnaire, a Robot Study questionnaire, and a Godspeed Questionnaire Series (GQS) [44] questionnaire. The Human Study questionnaire evaluates the perception of the participant from two perspectives: when the participant is one of the group members, and when the participant is the adjudicator. As a group member, the participant is asked questions relating to how polite the newcomer is and whether they liked the newcomer joining the group. As the newcomer, the participant is asked questions relating to how much they feel the people in the group wanted them to join it, and if they tried to find a comfortable approach path. The Robot Study questionnaire evaluates the perception of the participant when interacting with the robot as a group member. Questions are asked how polite participants thought the robot was in its approach behaviors, how sociable and human-like its behavior was, how much they liked when the robot joined the group and whether they preferred to play with humans or the robot. The GQS questionnaire, used to evaluate the perception of interactions with robots, measures 5 aspects of the robot, i.e., Anthropomorphism, Animacy, Likeability, Perceived Intelligence, Perceived Safety. Each aspect consists of questions assessed on a 1-5 Likert scale. Only the participants who took the robot-group interaction session answer the Robot Study and GQS post-study questionnaires.

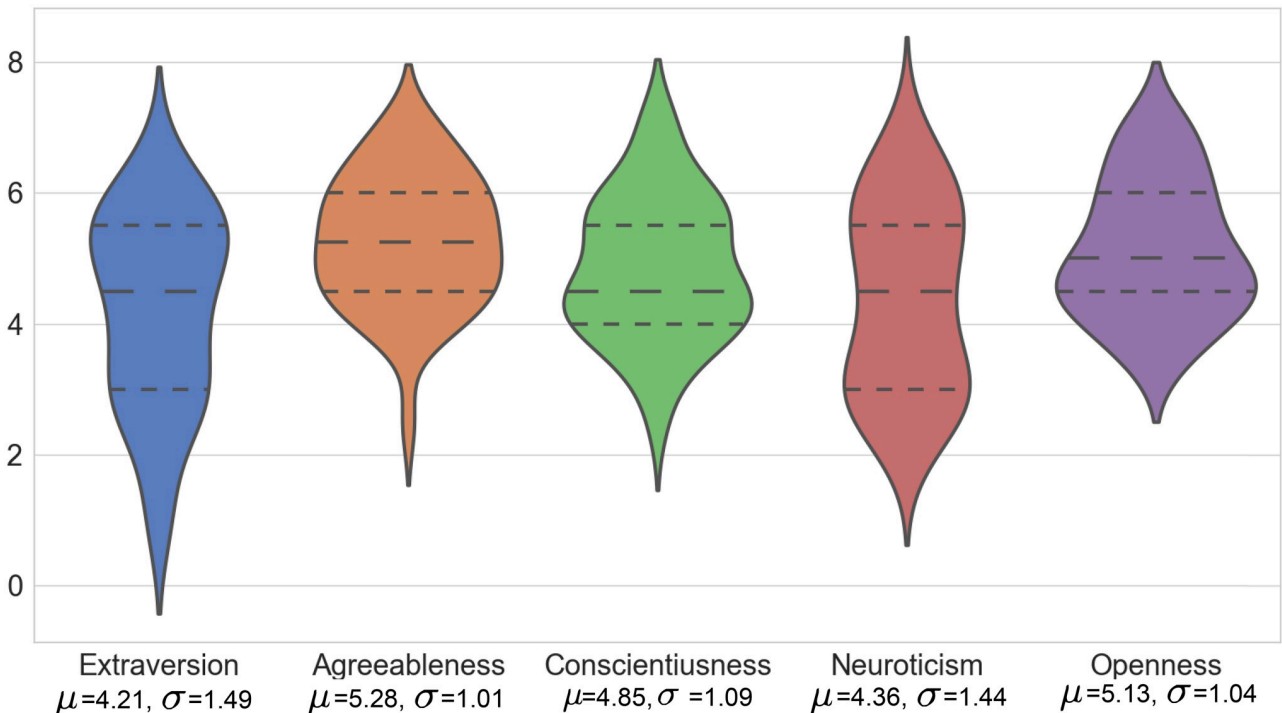

**Fig 12. The violin plot of the big-five personality traits across all participants.**

### Analysis of annotations and questionnaires

In the post-study questionnaires, the participants were asked about their perception of the newcomer (either a human player or a robot) via the questions *"How much do you like when the outside player joined your conversational group?"* from the Human Study questionnaire and the question *"How much do you like when the robot joined your conversational group?"* from the Robot Study questionnaire. We use *"Level of Accommodation"* (see Fig 13 left) to represent the answer to these questions. Wilcoxon rank-sum test gives p-value 0.028. Therefore at significance level 0.05, groups are more likely to accommodate a human newcomer than a robot newcomer. Accidentally, these questions are corresponding to the annotations that the groups adopt either *Accommodate* or *Ignore* behaviors. Similarly, we evaluate the ratio of behavior annotations (see Fig 13 right), i.e., the number of accommodation trials over all trials for each group. Wilcoxon rank-sum test gives p-value 0.002, which suggests that from the behavior aspect, groups are more likely to accommodate a human newcomer. The response to the question *"Do you prefer to play with human or robot?"* shows that the groups prefer to play the game with a human (M = 5.07, SD = 1.73).

While labeling the group behaviors, the researchers noticed that some groups are more likely to show accommodation behaviors. It is interesting to investigate if the accommodation behavior preference has a relation with a self-reported personality from the pre-study BFI-10 questionnaires. S2 Table. summarizes the self-reported personality and the percentage of labels. Since the role of newcomer is alternated between 4 participants, there is a potential that acquaintance may affect the accommodate behaviors made in the group. However, S2 Table shows that there does not appear to be any trend of increasing Accommodate behaviors as the experiments proceed. Averages are calculated for each pool, i.e., four participants, and Fig 14 shows the averages and the percentage of Accommodate labels. There is no similar correlation

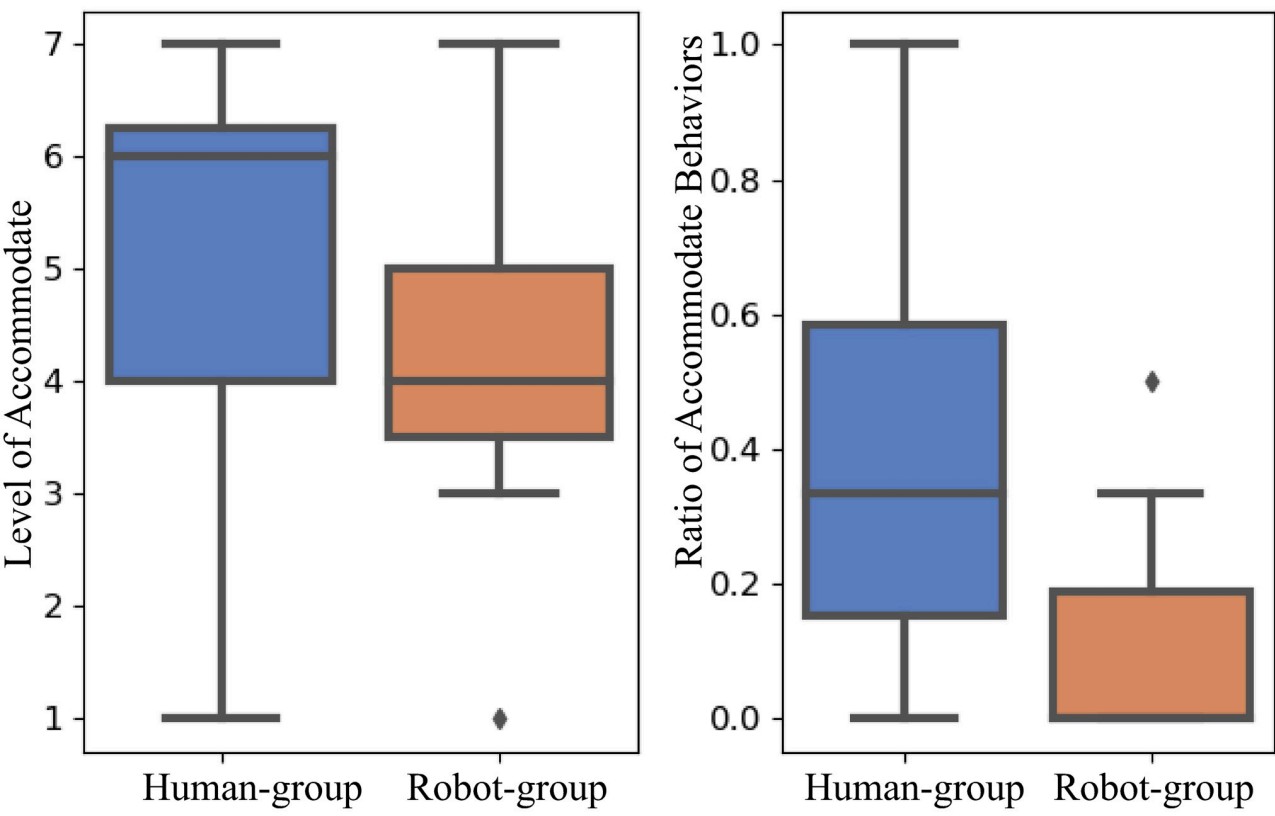

**Fig 13. The boxplot of the level of accommodation (left) and the ratio of accommodation behaviors (right).**

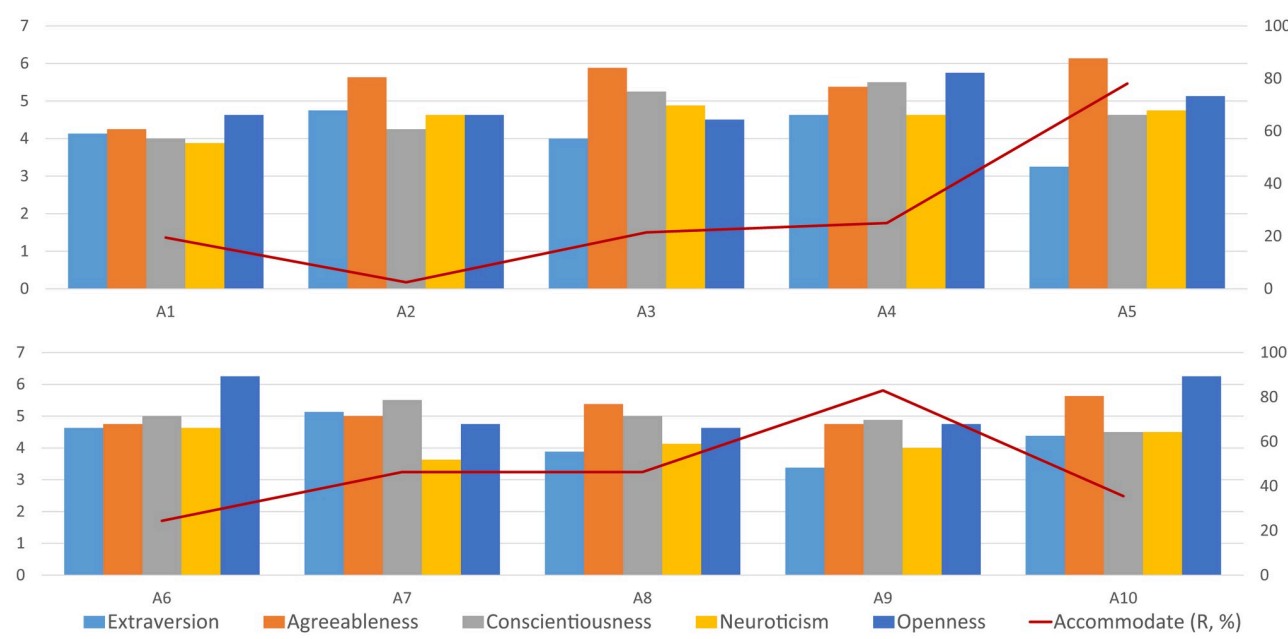

**Fig 14. The averaged personality of each pool (left axis) and the percentage of Accommodate labels (right axis).**

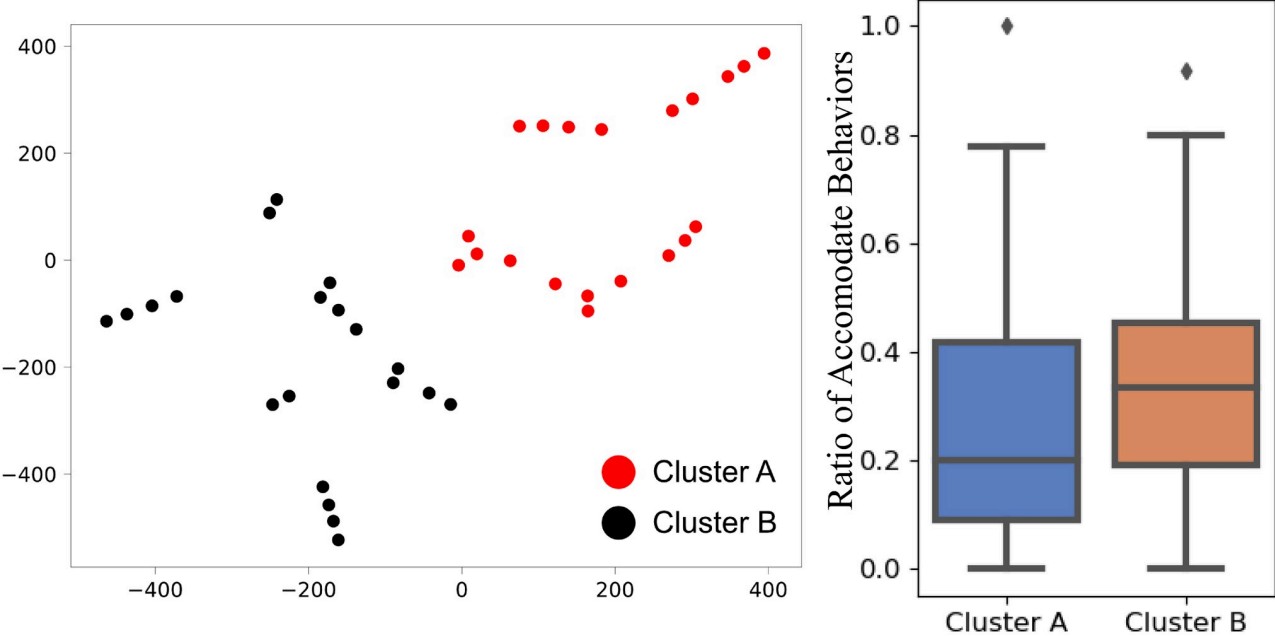

**Fig 15. The clustered group personalities on dimension-reduced data (left) and the boxplot of the ration of accommodate behaviors of two clusters (right).**

between any big-five personality dimension and the Accommodate behaviors. We thus combine all dimensions by clustering the groups to discover potential correlations. In order to cluster the groups into two classes, we collect the personality data of all three group members as features of one group. Then we use a t-SNE [46] algorithm to reduce the data dimension and k-means clustering to find clusters. Fig 15 shows all groups are clustered into two classes based on the personality data of group members. Wilcoxon rank-sum test gives p-value 0.12, which suggests that the self-reported personalities are not significantly related to the *Accommodate* or *Ignore* behaviors.

## Data protection and availability

The research and public dataset release has been approved by KTH's Data Protection Officer (DPO) to comply with Data Protection Regulation standards and ethics guidelines of the Swedish Ethical Review Authority, including informed consent from participants. The individuals depicted in the images in this manuscript have provided their informed consent (as outlined in PLOS consent form) to publish these case details. The CongreG8 dataset is free for research use, and updates will be made at https://zenodo.org/record/4537811. Queries can be addressed through contact with the corresponding authors.

## Use cases

The CongreG8 dataset have utility in a wide variety of domains, including the animation of embodied artificial characters, simulation of mobile robot behaviours and group behavior recognition. We present three use cases to demonstrate how the CongreG8 dataset has been used.

## Group behavior recognition

As previously mentioned in data annotation, when a newcomer approaches a conversational group, the group may dynamically react by adjusting their positions and orientations in order

to accommodate it. These reactions represent important cues to the newcomer about if and how they should plan their approach behaviors. The recognition and analysis of such socially-complaint dynamic group behaviors have rarely been studied in-depth and remain challenging in social multi-agent systems. We have developed novel neural networks trained on the CongreG8 dataset in recognizing such group behaviors [39]. Additionally, an online virtual chat-room is created to apply the group recognition model where a newcomer could get a real-time recognition of group behaviors (Fig 15 left and middle).

## Robot behavior generation

Robots that navigate to approach free-standing conversational groups should do so in a safe and socially-acceptable manner. However, it is challenging since it requires the robot to adopt socially acceptable paths in order not to make group members feel uncomfortable, e.g., due to violating their personal boundaries. Due to its importance of these approach behaviors for robots that have social roles, including mobile companion robots and delivery robots in social environments [47, 48], recently numerous methods and experiments have been done in deriving robot approach behaviors [49]. Mainly two ways have been considered to control robot trajectories in approaching small groups. The first one is to use computational models [22, 25, 50], and the second one is to use machine learning methods [4–6, 51]. The CongreG8 dataset can be used to support both methods. The computational methods rely on hand-crafted features to control the robot trajectories. Our dataset can be used either to derive features or to evaluate the computational model itself. On the other hand, due to the lack of datasets consisting of individuals approaching groups, the machine learning method either built upon prior computational models [6] or used synthetic datasets generated from the computational model for training [4]. We used human approaches group behaviors from the CongreG8 dataset to learn approach group behaviors based on generative adversarial imitation learning. The imitated behaviors are enabled in a Pepper robot to compare with robot behaviors generated from procedural models and WoZ [52]. Moreover, the CongreG8 dataset contains high-quality full-body motion data that can help in robot learning to learn human behaviors [53].

## Simulated group behaviors

Social behaviors and interactions are often integral to game-based learning environments, especially those involving social scenarios. A common requirement in such systems is to be able to embody behavior through animated and expressive virtual characters. Many game genres, such as role-playing games (RPGs) and real-time strategy games with crowds of characters, heavily rely on the ability to simulate group behaviors in order to maintain a sense of realism [54, 55]. However, creating such behaviors can be a time consuming and complex task requiring substantial technical expertise. Therefore, it either restricts the possibilities of what can be achieved or redirects the focus of game designers and pedagogists on technologies and away from the issues of creating engaging, educational experiences [56]. Besides, it is challenging to tackle complex situations such as forming conversation groups and recognizing social presence [57]. CongreG8 dataset offers full-body motion capture data that can be directly applied to create individuals' and groups' behaviors in the virtual environment (Fig 16 right). Thus, it allows more time to be devoted towards the design of game scenarios and accelerate and better use virtual characters in games, especially in social games with sophisticated virtual characters, while maintaining a high sense of realism.

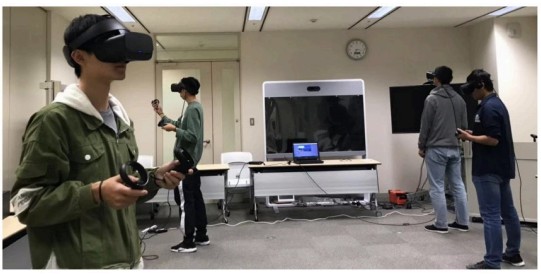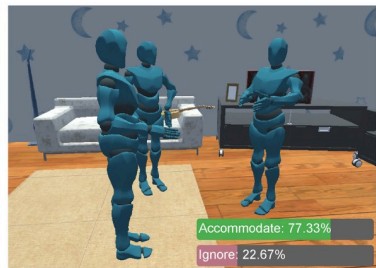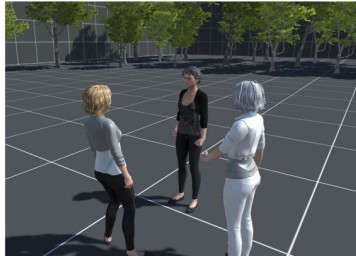

**Fig 16. Three example use cases of the CongreG8 dataset.** Group behavior recognition in an online virtual chatroom (left and middle) and simulated group behaviors in modeling group animations (right).

## Discussion

The group behaviors are currently labeled into two classes. However, the intensity of behaviors could vary, especially in the accommodate behaviors, e.g., the moving backward behavior is regarded as a much stronger accommodate behavior when compared to the eye-gaze behavior towards the newcomer. Thus HRI researchers could label multi-level classes of group behaviors via raw data from the CongreG8 dataset in the future. The CongreG8 dataset could potentially be coupled with trajectory prediction systems [4, 5, 58] as most of them use limited information such as locations and orientations. The CongreG8 dataset collects full-body data that represents not only location and orientation information, but also head orientation, upper-body behaviors, and full-body gestures. Machine learning models trained on our dataset could potentially offer a solution to enhance the perception capability of robots when moving into a group of people.

The number of collected robot-group trials is less than the human-group trials as the robot is controlled in a WoZ approach. We expect the human-group trials to be more necessary for the purpose of machine learning and robot imitation learning. However, the robot-group trials are valuable in the statistical analysis of robot versus human behaviors with group interactions. For example, the analysis of annotations and questionnaires show the group displays ignore behaviors more towards the robot newcomer than human newcomers. Our dataset contains both human-group and robot-group data that can be used to analyze the different behaviors when a human or a robot approaches the group. Leveraging the robot-group interaction data offers a start point to understand group behaviors when interacting with robots. It supports the generation of better robot behaviors and minimizes the difference between robot-group and human-group interactions.

From our observation on datasets containing free-standing conversation groups [7–9], groups with three and four members are the most dominant. Thus, we collect data from groups with three and four members considering time and resource limitations. CongreG8 dataset is updating by increasing its capacity, including collecting data of two-member groups and over-four-member groups. It also includes HRI data, where data-driven methods generate the robot trajectories. Furthermore, the CongreG8 dataset shows its capability by training data-driven models in behavior recognition [39] and robot joining group behaviors [52]. It paves the way for applying these models in data with varying group configurations.

When collecting HRI data, a WoZ approach was chosen since there is no good automatic control system for moving a robot into an appropriate position in a dynamic group situation by taking full-body behaviors as inputs. CongreG8's main purpose is to provide data of human group approach interactions as a basis for training machine learning models for robots approaching groups of humans in a socially compliant manner and then to replace the WoZ

control [52]. Generates robot approaching behaviors by imitating Human-Human Interaction data from CongreG8, and the HRI data where the robot is controlled by data-driven models is included in the CongreG8 dataset.

## Conclusion

We presented the CongreG8 dataset, a novel dataset of human/robot-group interaction data. CongreG8 is the first dataset of its kind in the literature. We expect it will play a significant role in promoting standardization in HRI research involving approach behaviors in groups of humans and robots. The CongreG8 dataset also contains a large human-group interaction data for training models of group dynamics, including behavior recognition, behavior generation, or personality interpretation. We thus expect the dataset will also help to build artificial systems with group dynamics.

## Supporting information

**S1 Table. Overview of the CongreG8 dataset.**
(PDF)

**S2 Table. The big-five personality traits (E, A, C, N, O) and group behavior labels (Accommodate percentage A% and Ignore percentage I%) across all groups.** Rows labelled "Average 1"-"Average 10" represent average values for each pool of four participants respectively.
(PDF)

## Acknowledgments

The authors warmly thank the PMIL motion capture lab at KTH for their help with data collection.

## Author Contributions

**Conceptualization:** Fangkai Yang.

**Data curation:** Fangkai Yang, Yuan Gao, Ruiyang Ma, Sahba Zojaji.

**Formal analysis:** Fangkai Yang.

**Funding acquisition:** Christopher Peters.

**Investigation:** Fangkai Yang.

**Methodology:** Fangkai Yang.

**Project administration:** Fangkai Yang.

**Resources:** Fangkai Yang, Yuan Gao, Ginevra Castellano, Christopher Peters.

**Software:** Fangkai Yang.

**Supervision:** Fangkai Yang, Ginevra Castellano, Christopher Peters.

**Validation:** Fangkai Yang.

**Visualization:** Fangkai Yang.

**Writing – original draft:** Fangkai Yang.

**Writing – review & editing:** Fangkai Yang, Yuan Gao, Sahba Zojaji, Ginevra Castellano, Christopher Peters.

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
