## [Decision Letter · Decision Letter 0]

9 Sep 2020

PONE-D-20-21440

A Dataset of Human and Robot Approach Behaviors into Small Free-Standing Conversational Groups

PLOS ONE

Dear Dr. Yang,

Thank you for submitting your manuscript to PLOS ONE. After careful consideration, we feel that it has merit but does not fully meet PLOS ONE’s publication criteria as it currently stands. Therefore, we invite you to submit a revised version of the manuscript that addresses the points raised during the review process.

We look forward to receiving your revised manuscript.

Kind regards,

Josh Bongard

Academic Editor

PLOS ONE

Journal Requirements:

2. We note that Figures [1,4,5] includes an image of a [patient / participant / in the study]. 

Reviewers' comments:

Reviewer's Responses to Questions

**Comments to the Author**

1. Is the manuscript technically sound, and do the data support the conclusions?

Reviewer #1: Yes

Reviewer #2: Yes

2. Has the statistical analysis been performed appropriately and rigorously? 

Reviewer #1: Yes

Reviewer #2: Yes

3. Have the authors made all data underlying the findings in their manuscript fully available?

Reviewer #1: Yes

Reviewer #2: Yes

4. Is the manuscript presented in an intelligible fashion and written in standard English?

Reviewer #1: Yes

Reviewer #2: Yes

5. Review Comments to the Author

Reviewer #1: In this paper, a novel dataset has been presented to model group dynamics during conversations, in particular, approaching behaviour. The dataset comprises the recordings of interactions between humans, but also between humans and a real humanoid robot. The data collection and annotation have been conducted in a systematic way with clear research questions. The paper also draws an implication that how humans see a human newcomer is different from how humans see a robot newcomer. If the authors make their dataset publicly available, it will be beneficial to the research community.

MAJOR COMMENTS

Although it is original, the dataset has several limitations: The dataset focuses on a fixed group configuration (4 participants only) and a specific interaction scenario/a role-play setting. It is not clear how algorithms trained such a controlled and limited dataset can be scaled to practical applications with varying conversational group configurations, especially for the use cases discussed in the paper.

The number of participants/groups is limited (40 participants/10 groups only). Approaching behaviour has been captured for the same group several times, which introduces its own biases as after a couple interactions the participants will already become acquainted. It is not clear from the paper what is aimed to be captured. If they aim to capture the tendency of a group of people to accept a newcomer, is the designed experiment appropriate for this aim?

The paper does not provide further insight into why humans prefer a human newcomer over a robot newcomer. This might be due to the robot's limited capabilities for maintaining interaction during the game. The description of the robot control is very brief in the paper, and it is not clear what behaviour is automatic/what behaviour is controlled by the human operator.

The dataset mainly focuses on motion capture data. What are practicalities of motion capture data in real-life scenarios, where it is not possible to ask humans to wear mobcap suits? Also, how can this approach be implemented on a robot to perceive and approach a group?

The authors should include a section summarising the data statistics. It would be helpful for the reader to see some visualisations of how group use the space, how much they move, how they move when a newcomer joins the group, etc. Further statistics regarding the distribution of labels in particular, personality, accommodate and ignore should be presented in the paper.

Taken together, the paper needs a more detailed analysis of the collected data and labels and a discussion about the limitations.

MINOR COMMENT

The authors should discuss the following datasets/papers in the related work section:

• JRDB: A Dataset and Benchmark for Visual Perception for Navigation in Human Environments

• Robot-centric perception of human groups

• RICA: Robocentric Indoor Crowd Analysis Dataset

• Robocentric Conversational Group Discovery

Reviewer #2: Summary of the paper:

The paper describes a new publicly available dataset of motion capture recordings of free standing, conversing groups of 3 people that accommodate either a new person or a teleoperated robot.

Furthermore, psychological and HRI questionnaires are collected from the participants.

This is the first dataset of its kind and can be used for different scientific research in the future as pointed out by 3 use-cases that are described in the paper.

The paper focuses on the description of the dataset and its collection procedure.

The analysis of the data showed that that people prefer more to accommodate other people rather than robots to join their group.

--

Review opinion:

The paper is written in a clear and understandable manner.

The dataset and its collection procedure are described in detail.

The dataset is of scientific important as it is the first that collects motion data of the joining behavior of a human to a free-standing group of humans in such detail.

The related work lists the relevant literature.

There are a few papers that the authors could add to the "Group interaction research" section if they believe it adds to their paper:

Livramento, R., Avelino, J., & Moreno, P. (2020, April). Natural Data-driven Approaching Behaviors of Humanoid Mobile Robots for F-Formations. In 2020 IEEE International Conference on Autonomous Robot Systems and Competitions (ICARSC) (pp. 338-344). IEEE.

Pathi, S. K., Kristofferson, A., Kiselev, A., & Loutfi, A. (2019, October). Estimating Optimal Placement for a Robot in Social Group Interaction. In 2019 28th IEEE International Conference on Robot and Human Interactive Communication (RO-MAN) (pp. 1-8). IEEE.

Escobedo, A., Spalanzani, A., & Laugier, C. (2014, September). Using social cues to estimate possible destinations when driving a robotic wheelchair. In 2014 IEEE/RSJ International Conference on Intelligent Robots and Systems (pp. 3299-3304). IEEE.

Rios-Martinez, J., Spalanzani, A., & Laugier, C. (2011, September). Understanding human interaction for probabilistic autonomous navigation using Risk-RRT approach. In 2011 IEEE/RSJ International Conference on Intelligent Robots and Systems (pp. 2014-2019). IEEE.

The paper focuses on the documentation and description of the dataset.

Its major finding is that humans prefer more to accommodate other humans than robots.

It lists other papers in its use-cases that used the dataset for further scientific studies (see ref 32, 45), but does not provide a further detailed analysis of the data itself.

In the context of publishing the dataset as a scientific paper, I would encourage the authors to add further analysis of the collected data, for example as described in line 400:

"the CongreG8 dataset could be analyzed further to discover the relation between the self-reported personality and actually performed behaviors in groups".

I gave therefore as recommendation: "Major Revision".

Nonetheless, I believe it is up to the editor to decide if the scientific contribution in terms of the analysis of the collected data is enough to grant a publication at PlosOne.

If the editor accepts this, then my recommendation is "Accept".

--

Minor points:

368 - Not clear what this sentence wants to say.

Besides, complex situations such as forming conversation groups and recognizing each other[s] presence need to be tackled [with a few challenges ??]

Table 1, Page 7: use bullet points also for "Tutorial" and "Break" explanation, similar to "Robot-Group-Interaction" section.

I believe the paper could be shortened by reducing details or shortening descriptions about the dataset format:

- Figure 1 and 2 could be combined

- Figure 4 and 5 could be combined

- Figure 7 could be removed

- Figure 8 could be removed

- details about file formats could be removed in paragraph 230 - 239

Generally, detailed points about the used software to collect data or fileformats can be mentioned in the online documentation of the publicly available dataset.

--

Grammatical errors:

(Please note that I am not a English native speaker.)

189 - In the Greetings stage ...

192 - In the meanwhile, a/the face tracker is ...

397 - It helps [to] generate better robot behaviors and [to] minimize the difference between robot-group and human-group interactions.

6. PLOS authors have the option to publish the peer review history of their article (what does this mean?). If published, this will include your full peer review and any attached files.

Reviewer #1: No

Reviewer #2: No

---

## [Author Response · Author response to Decision Letter 0]

21 Dec 2020

The responses are covered in one of uploaded files, i.e., 'Response to Reviewers'.

---

## [Decision Letter · Decision Letter 1]

8 Feb 2021

A Dataset of Human and Robot Approach Behaviors into Small Free-Standing Conversational Groups

PONE-D-20-21440R1

Dear Dr. Yang,

We’re pleased to inform you that your manuscript has been judged scientifically suitable for publication and will be formally accepted for publication once it meets all outstanding technical requirements.

Kind regards,

Josh Bongard

Academic Editor

PLOS ONE

Additional Editor Comments (optional):

Reviewers' comments:

Reviewer's Responses to Questions

**Comments to the Author**

1. If the authors have adequately addressed your comments raised in a previous round of review and you feel that this manuscript is now acceptable for publication, you may indicate that here to bypass the “Comments to the Author” section, enter your conflict of interest statement in the “Confidential to Editor” section, and submit your "Accept" recommendation.

Reviewer #1: All comments have been addressed

Reviewer #2: (No Response)

2. Is the manuscript technically sound, and do the data support the conclusions?

Reviewer #1: Yes

Reviewer #2: Yes

3. Has the statistical analysis been performed appropriately and rigorously? 

Reviewer #1: Yes

Reviewer #2: Yes

4. Have the authors made all data underlying the findings in their manuscript fully available?

Reviewer #1: Yes

Reviewer #2: Yes

5. Is the manuscript presented in an intelligible fashion and written in standard English?

Reviewer #1: Yes

Reviewer #2: Yes

6. Review Comments to the Author

Reviewer #1: Overall, this paper introduced a novel dataset and a systematic study of behaviours of free-standing groups. The dataset has been made available already, and it will be beneficial to the HHI and HRI community from multiple aspects.

The authors incorporated the comments by the reviewers adequately. In particular, they included a summary of the data statistics and extended the discussion section by considering the limitations of their work. However, discussion section could be structured better by presenting limitations separately and then summarising the solutions. [52] already used this dataset, for instance, the achievements and the potential applications of this dataset could be emphasised better.

Reviewer #2: Summary of review R1:

The authors have responded to several points noted by the reviewers.

Additional literature for "Group interaction research" and references to other Datasets were added.

More information regarding the participants of the study were added.

New Figures and Tables were added to provide more information and statistics of the collected data:

Fig 10. (a) shows a heatmap of the participants overall position and (b,c) examples of "Ignore" and "Accept" behaviors.

Fig. 14 shows the mean personality traits per participant group and their "Acceptance" rate.

Table S1: Summarized information about the dataset.

Table S2: Detailed statistics of the participants personality traits.

--

Review opinion:

Although some new statistics were added to the paper, my main comment from the initial review still stands:

The paper focuses on the documentation and description of the dataset, which is also implied by the title of the paper.

In terms of data analysis, its major finding is that humans accommodate other humans more than robots which could not be related to the measured personality traits.

It lists other papers in its use-cases that used the dataset for further scientific studies, but does not provide a detailed analysis of the data itself.

Many aspects of the collected data were not considered for the analysis, for example the 3D body positions.

In the context of publishing the dataset as a scientific paper, I would encourage the authors to focus the paper stronger on the data analysis than a description of their dataset.

I gave therefore as recommendation: "Major Revision".

Nonetheless, I believe it is up to the editor to decide if the scientific contribution in terms of collecting, describing and the publication of such a dataset is enough to grant a publication at PlosOne.

If the editor accepts this, then my recommendation is "Accept".

--

Minor points:

Fig. 10 (a) - It is not described how the heatmap was created, i.e. if it is for a single trial or over many trials, and if the positions were maybe preprocessed, for example to be relative to the group center.

31 - "Those existing datasets that contain free-standing groups [7{9] have a limited number of samples of individuals approaching the group and typically contain only 2D location information, making it difficult to train neural networks."

The sentence seems to imply that networks are difficult to train because the data is 2D and there is a limited number of samples. I guess it should only say that it is difficult to train because of a limited number of samples.

I do not see why 2D data makes it hard to train a network.

142-143 - "robot-group condition, in which a human plays the adjudicator/newcomer role".

I guess it should be a robot in this condition who plays the adjudicator role and not a human.

--

Grammatical errors:

196 - "in Section ."  "in Section."

7. PLOS authors have the option to publish the peer review history of their article (what does this mean?). If published, this will include your full peer review and any attached files.

Reviewer #1: **Yes: **Oya Celiktutan

Reviewer #2: No

---

## [Editor Report · Acceptance letter]

17 Feb 2021

PONE-D-20-21440R1 

A Dataset of Human and Robot Approach Behaviors into Small Free-Standing Conversational Groups 

Dear Dr. Yang:

I'm pleased to inform you that your manuscript has been deemed suitable for publication in PLOS ONE. Congratulations! Your manuscript is now with our production department. 

Kind regards, 

on behalf of

Dr. Josh Bongard 

Academic Editor

PLOS ONE